

# Risk of acute exacerbation between acetaminophen and ibuprofen in children with asthma

Lin-Shien Fu[1,2], Che-Chen Lin[3], Chia-Yi Wei[3], Ching-Heng Lin[3] and Yung-Chieh Huang[1,4]

[1] Department of Pediatrics, Taichung Veterans General Hospital, Taichung, Taiwan
[2] Department of Pediatrics, National Yang-Ming University, Taipei, Taiwan
[3] Department of Medical Research, Taichung Veterans General Hospital, Taichung, Taiwan
[4] Division of Pediatrics, Puli Branch, Taichung Veterans General Hospital, Nantou, Taiwan

## ABSTRACT

**Background:** Antipyretics are widely prescribed in pediatric practice. Some reports have mentioned that acetaminophen and non-steroid anti-inflammatory drugs may negatively affect asthma control by causing asthma exacerbation (AE). However, many confounding factors can also influence the risks. We assessed the impact of using acetaminophen or ibuprofen on AE in asthmatic children, especially those with strong risk factors.

**Methods:** We used the 2010 Taiwan National Health Insurance Research Database and identified 983 children with persistent asthma aged 1–5 years old; among them, 591 used acetaminophen alone and 392 used ibuprofen alone in 2010. Then, we analyzed the risk of AE over 52 weeks in the patients with and without severe AE in the previous year.

**Results:** The ibuprofen group had a higher risk of an emergency room (ER) visit or hospitalization for AE (odds ratio (OR) = 2.10, 95% confidence interval (CI) [1.17–3.76], $P = 0.01$). Among asthmatic children who had severe AE in the previous year, the risk of AE was higher in the ibuprofen group than in the acetaminophen group (OR = 3.28, 95% CI [1.30–8.29], $P = 0.01$), where as among those who did not, the risks of AE were similar between the acetaminophen and ibuprofen groups (OR = 1.52, 95% CI [0.71–3.25], $P = 0.28$).

**Conclusions:** Among young asthmatic children, use of ibuprofen was associated with a higher risk of AE than acetaminophen, if they had severe AE with ER visit or hospitalization in the previous year. Pediatricians should use antipyretics among children with asthma after a full evaluation of the risk.

## INTRODUCTION

Acetaminophen and ibuprofen are the most widely used medications in children, the major indication of their use being fever. Studies have suggested that the risk of asthma and wheezing increases after acetaminophen use in several situations, including prenatal administration, administration in infants, and exposure in the previous year

Corresponding authors
Ching-Heng Lin, epid@vghtc.gov.tw
Yung-Chieh Huang,
huang1985john@yahoo.com.tw

(*Bacharier et al., 2008*; *Gonzalez-Barcala et al., 2013*; *Henderson & Shaheen, 2013*; *Kreiner-Moller et al., 2012*; *Muc, Padez & Pinto, 2013*; *Scialli et al., 2010*; *Shaheen et al., 2000*; *Thiele et al., 2013*; *Wong et al., 2007*). However, none of these studies have reported on ibuprofen use, even though it is another frequently used antipyretic. Furthermore, these studies did not rule out the confounding effect of concurrent respiratory tract infection (RTI), which may play a more important role in asthma than acetaminophen use (*Schnabel & Heinrich 2010*; *Sordillo et al., 2015*). Studies assessing the influence of acetaminophen on lung function have shown conflicting results in adults and children (*Ioannides et al., 2014*; *McKeever et al., 2005*; *Soferman et al., 2013*).

Studies comparing acetaminophen and ibuprofen for asthma control have also revealed inconsistent results. Short-term use of acetaminophen for febrile illness in children aged 6 months to 12 years with asthma did not lead to more hospitalizations but did result in more unscheduled outpatient department (OPD) visits for asthma exacerbation (AE) than ibuprofen use (*Lesko et al., 2002*). However, after comparing the as-needed use of acetaminophen and ibuprofen for 48 weeks, a recent double-blind randomized controlled study revealed no difference in asthma control or asthma treatment with systemic corticosteroids in children aged 1–5 years with mild persistent asthma (*Sheehan et al., 2016*). *Matok et al. (2017)* examined the association between antipyretics and wheezing in children with febrile illness, and found that ibuprofen was associated with a lower risk of wheezing.

According to reports from the Global Initiative for Asthma, severe AE in the past year is the strongest independent predictor of AE (*Asthma GIf, 2017*). AE is defined as an acute or sub-acute episode of a progressive increase in asthma symptoms, associated with airflow obstruction; the most commonly examined exacerbation outcomes are the need for systemic corticosteroids and urgent unscheduled asthma-related care, specifically emergency room (ER) visits, hospitalization, or unscheduled OPD visits (*Papadopoulos et al., 2012*). In this study, we used the Taiwan National Health Insurance Research Database (NHIRD) to compare the risk of AE between 1-year use of acetaminophen and 1-year use of ibuprofen in children aged 1–5 years with persistent asthma, focusing on the risk factors, including severe AE in the past year.

## PATIENTS AND METHODS

### Data source

Taiwan established the National Health Insurance program in 1995, and more than 98% of Taiwan residents were insured under this system. The NHIRD was provided by the National Health Research Institutes of Taiwan. The diagnostic codes in the database were based on the International Classification of Diseases, Ninth Revision, Clinical Modification (ICD-9-CM). In this study, we used the 2010 NHIRD, which contained the original claims data of 1,000,000 beneficiaries randomly sampled from the whole population.

### Study design

Figure 1 presents a flowchart of patient inclusion in the study. There were 4,444 pediatric asthma cases in the database on January 1, 2010. The diagnosis of asthma was based on

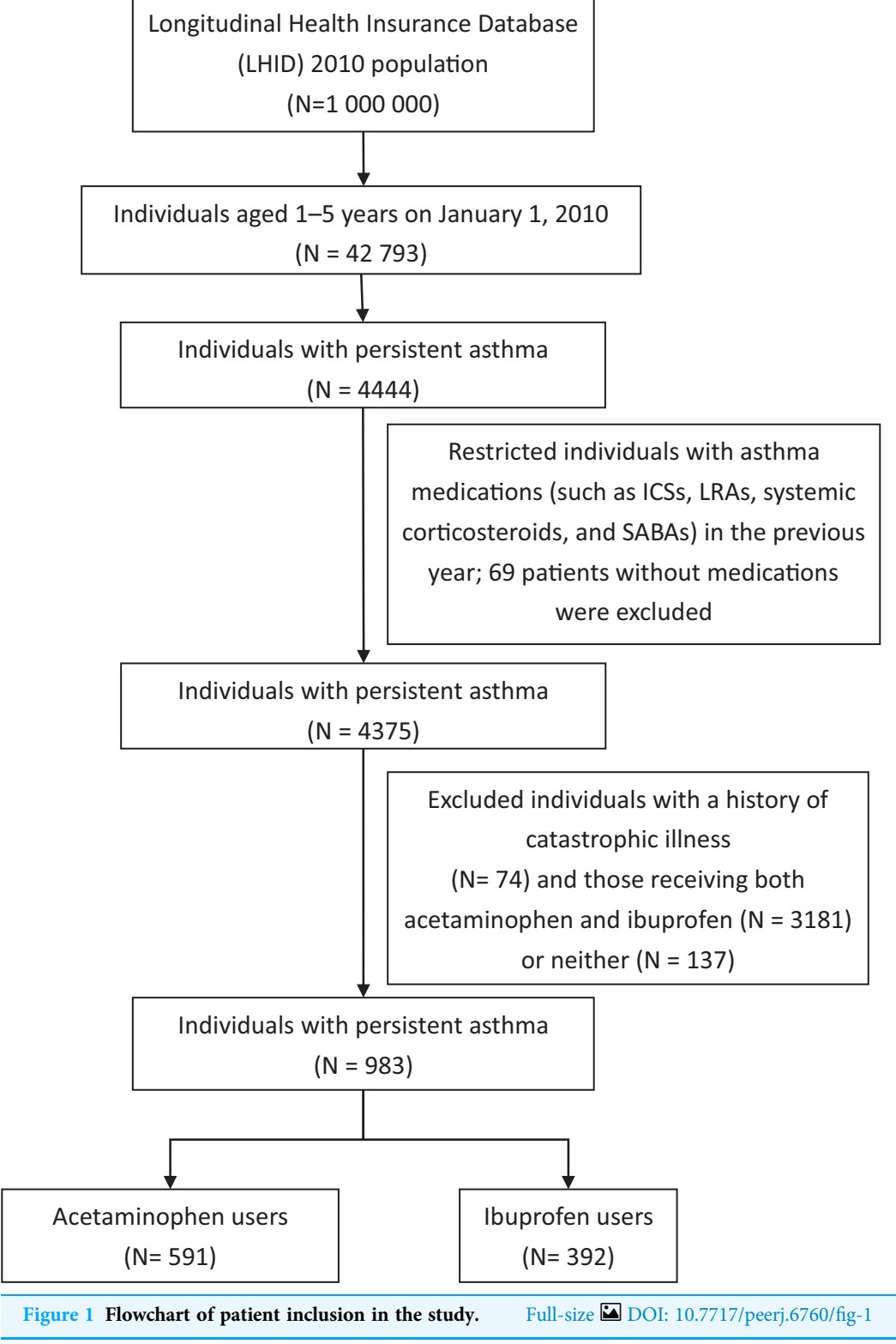

**Figure 1  Flowchart of patient inclusion in the study.**

ICD-9-CM 493.X in at least one ER visit, one hospital admission, or three OPD visits, and the asthma drugs used included inhaled corticosteroids (ICSs), leukotriene receptor antagonists (LTRAs), systemic corticosteroids, and short-acting β-agonists.

The primary endpoint of this study was development of AE, which was identified on the basis of an ER visit or hospital admission with a primary ICD-9-CM code of 493.X within 1 year. We considered AE and systemic corticosteroid treatment for asthma in the previous year as risk factors in this study.

This study was approved by the institutional review board of Taichung Veterans General Hospital (No. CE16220B). For the protection of privacy, the identities of the patients, physicians, and institutions were scrambled in accordance with the Personal Electronic Data Protection Law.

## Statistical analysis

We evaluated the baseline characteristics of the patients with asthma, including age, sex, age at asthma onset, asthma status, and medication status in the year preceding the baseline. Continuous variables were expressed as the mean ± standard deviation, and the difference between two groups was assessed using the $t$-test. Categorical variables were calculated, and differences in the distribution were assessed using the chi-square test. To compare the risks of AE between acetaminophen and ibuprofen users, the odds ratios (ORs) and corresponding 95% confidence intervals (CIs) were estimated by using univariate and multivariate logistic regression models. We also measured the cumulative incidence of AE in acetaminophen and ibuprofen users by using the Kaplan–Meier method and tested differences in the curves by using the log-rank test. The incidence curves were drawn using R software (R Core Team, 2018), and data management and statistical analyses were performed using SAS 9.4 (SAS Institute, Cary, NC, USA). A two-sided $P$-value of <0.05 was set as the significance level.

## RESULTS

Table 1 summarizes the demographic characteristics of the two groups. After excluding patients who had no asthma-related prescription ($n = 69$), those with severe systemic diseases ($n = 74$), mixed antipyretic users ($n = 3,181$), and those with no acetaminophen or ibuprofen prescription ($n = 137$), 591 children with acetaminophen prescription alone and 392 with ibuprofen prescription alone remained. The mean age of the patients in the ibuprofen group was lower than that in the acetaminophen group ($4.0 ± 0.9$ vs. $4.2 ± 0.9$ $y$, $P = 0.008$), and the ibuprofen group had a higher percentage of boys (67.1% vs. 59.7%, $P = 0.02$). There were no between-group differences in age of asthma onset, severe AE (ER visit or hospitalization), frequency of upper RTI, or asthma-related prescriptions including ICSs, LTRAs, and systemic corticosteroids.

We first analyzed some possible risk factors for AE after adjusting for age, sex, and patient asthma and medication status in the previous year (Table 2). An ER visit or hospitalization in the previous year was associated with a high risk (OR = 10.6,

**Table 1 Baseline characteristics of subgroups of patients with asthma.**

| Variables | Acetaminophen users $N = 591$ | Ibuprofen users $N = 392$ | P-value |
|---|---|---|---|
| Age, year (SD) | 4.2 (0.9) | 4.0 (0.9) | 0.008 |
| Sex | | | 0.02 |
|    Female (%) | 238 (40.3) | 129 (32.9) | |
|    Male (%) | 353 (59.7) | 263 (67.1) | |
| Age at onset of asthma, month (SD) | 26.5 (13.2) | 25.3 (13.3) | 0.15 |
| Asthma exacerbation status in the previous year (%) | 71 (12.0) | 37 (9.4) | 0.21 |
|    Emergency visits for asthma (%) | 37 (6.3) | 18 (4.6) | 0.27 |
|    Hospitalizations for asthma (%) | 43 (7.3) | 23 (5.9) | 0.39 |
| Medication status in the previous year | | | |
|    Use of ICS (%) | 116 (19.6) | 61 (15.6) | 0.10 |
|    Use of leukotriene receptor antagonist (%) | 150 (25.4) | 115 (29.3) | 0.17 |
|    Use of systemic corticosteroids (%) | 267 (45.2) | 156 (39.8) | 0.10 |
| Diagnosed upper respiratory tract infection | | | |
|    In the previous year (%) | 589 (99.7) | 388 (99.0) | 0.18 |
|    In 2010 (%) | 582 (98.5) | 378 (96.4) | 0.04 |

Note:
  SD, standard deviation; ICS, inhaled corticosteroids.

**Table 2 Analysis of risk factors for AE.**

| Variables | Asthma exacerbation | | Adjusted OR | |
|---|---|---|---|---|
| | No (%) | Yes (%) | (95% CI) | P-value |
| Antipyretics[*] | | | | |
|    Acetaminophen | 564 (60.9) | 27 (47.4) | Reference | |
|    Ibuprofen | 362 (39.1) | 30 (52.6) | 2.10 [1.17–3.76] | 0.01 |
| Age, year (SD) | 4.1 (0.9) | 3.8 (0.9) | 0.80 [0.60–1.07] | 0.13 |
| Sex | | | | |
|    Female | 346 (37.4) | 21 (36.8) | Reference | |
|    Male | 580 (62.6) | 36 (63.2) | 1.03 [0.57–1.86] | 0.93 |
| Patient status in the previous year | | | | |
|    Asthma AE[‡] | | | | |
|       No | 846 (91.4) | 29 (50.9) | Reference | |
|       Yes | 80 (8.6) | 28 (49.1) | 10.6 [5.90–18.9] | <0.0001 |
|    Use of systemic corticosteroids[#] | | | | |
|       No | 547 (59.1) | 13 (22.8) | Reference | |
|       Yes | 379 (40.9) | 44 (77.2) | 3.07 [1.56–6.05] | 0.001 |

Notes:
  [*] Adjusted for age, sex, and patient status in the previous year (AE and use of systemic corticosteroids).
  [‡] Adjusted for age, sex, and AE in the previous year.
  [#] Adjusted for age, sex, and use of systemic corticosteroids in the previous year.
  OR, odds ratios; CI, confidence intervals.

95% CI [5.90–18.9], $P < 0.0001$). In addition, the ibuprofen group had a higher risk of AE than did the acetaminophen group after adjustment for sex, age, and patient status in the previous year (OR = 2.10, 95% CI [1.17–3.16], $P = 0.01$). The ibuprofen group

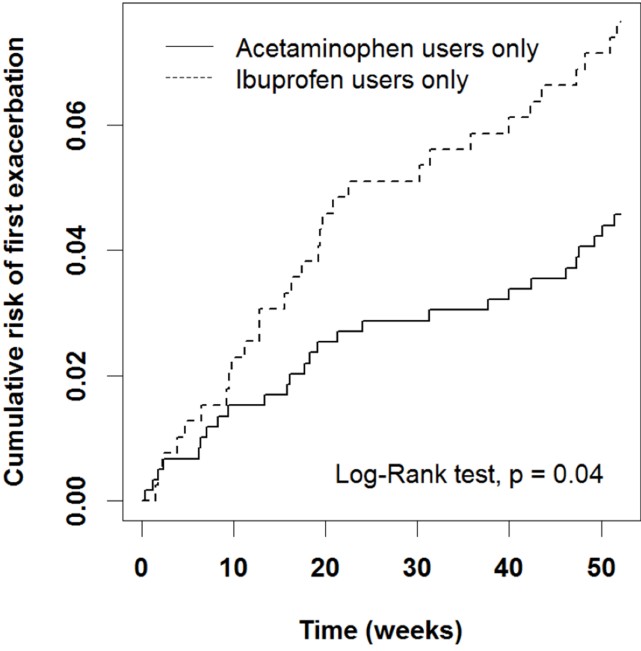

**Figure 2 Cumulative risk of first AE in acetaminophen and ibuprofen users.** The ibuprofen group exhibited a significantly higher cumulative risk of AE from early weeks to the end of the study period.

exhibited a significantly higher cumulative risk of AE from early weeks to the end of the study period (Fig. 2, $P = 0.04$).

We then compared the risk of AE between patients in the acetaminophen and ibuprofen groups who had systemic corticosteroid prescription and an asthma-related ER visit or hospitalization in the previous year (Table 3A). Among children who had an asthma-related ER visit or hospitalization in the previous year, those in the ibuprofen group had a higher risk of AE than did those in the acetaminophen group (OR = 3.28, 95% CI [1.30–8.29], $P = 0.01$). In children with no asthma-related ER visitor or hospitalization in the previous year, the risks of AE in both the acetaminophen and ibuprofen groups were similar ($P = 0.28$). For children who had systemic corticosteroid prescription in the previous year, the ibuprofen group had a higher risk of AE than the acetaminophen group (OR = 2.30, 95% CI [1.61–4.54], $P = 0.02$). When we further analyzed our results according to the patients' sex, we found that among female children (Table 3B) with asthma who had an asthma-related ER visit or hospitalization in the previous year, those in the ibuprofen group had a higher risk of AE than did those in the acetaminophen group (OR = 5.15, 95% CI [1.03–25.7], $P = 0.05$). Among female children who had systemic corticosteroid prescription in the previous year, the ibuprofen group had a higher risk of AE than the acetaminophen group (OR = 3.6, 95% CI [1.14–11.43], $P = 0.03$). However, there were no significant findings among male children with asthma (Table 3C).

When we subdivided systemic corticosteroid prescription into OPD visits and ER visits or hospitalization for asthma (Table 4), children receiving systemic corticosteroids in the OPD showed similar risks of AE in the acetaminophen and ibuprofen groups

**Table 3 Risk of acetaminophen and ibuprofen use in subgroups of children, female children, and male children with asthma.**

**(A) Risk of acetaminophen and ibuprofen use in subgroups of children with asthma**

| Antipyretics | Without asthma exacerbation in the previous year | | | | Asthma exacerbation in the previous year | | | |
|---|---|---|---|---|---|---|---|---|
| | Asthma exacerbation | | Adjusted OR | | Asthma exacerbation | | Adjusted OR | |
| | No (%) | Yes (%) | (95% CI) | P-value | No (%) | Yes (%) | (95% CI) | P-value |
| Acetaminophen | 506 (97.3) | 14 (2.7) | Reference | | 58 (81.7) | 13 (18.3) | Reference | |
| Ibuprofen | 340 (95.8) | 15 (4.2) | 1.52 [0.71–3.25] | 0.28 | 22 (59.5) | 15 (40.5) | 3.28 [1.30–8.29] | 0.01 |

| Antipyretics | Non-use of systemic corticosteroids in the previous year | | | | Use of systemic corticosteroids in the previous year | | | |
|---|---|---|---|---|---|---|---|---|
| | Asthma exacerbation | | Adjusted OR | | Asthma exacerbation | | Adjusted OR | |
| | No (%) | Yes (%) | (95% CI) | P-value | No (%) | Yes (%) | (95% CI) | P-value |
| Acetaminophen | 318 (98.1) | 6 (1.9) | Reference | | 246 (92.1) | 21 (7.9) | Reference | |
| Ibuprofen | 229 (97.0) | 7 (3.0) | 1.71 [0.56–5.29] | 0.35 | 133 (85.3) | 23 (14.7) | 2.30 [1.16–4.54] | 0.02 |

**(B) Risk of acetaminophen and ibuprofen use in subgroups of female children with asthma**

| Antipyretics | Without asthma exacerbation in the previous year | | | | Asthma exacerbation in the previous year | | | |
|---|---|---|---|---|---|---|---|---|
| | Asthma exacerbation | | Adjusted OR | | Asthma exacerbation | | Adjusted OR | |
| | No (%) | Yes (%) | (95% CI) | P-value | No (%) | Yes (%) | (95% CI) | P-value |
| Acetaminophen | 204 (65.0) | 7 (63.6) | Reference | | 24 (75.0) | 3 (30.0) | Reference | |
| Ibuprofen | 110 (35.0) | 4 (36.4) | 1.01 [0.28–3.66] | 0.98 | 8 (25.0) | 7 (70.0) | 5.15 [1.03–25.7] | 0.05 |

| Antipyretics | Non-use of systemic corticosteroids in the previous year | | | | Use of systemic corticosteroids in the previous year | | | |
|---|---|---|---|---|---|---|---|---|
| | Asthma exacerbation | | Adjusted OR | | Asthma exacerbation | | Adjusted OR | |
| | No (%) | Yes (%) | (95% CI) | P-value | No (%) | Yes (%) | (95% CI) | P-value |
| Acetaminophen | 130 (62.8) | 4 (80.0) | Reference | | 98 (70.5) | 6 (37.5) | Reference | |
| Ibuprofen | 77 (37.2) | 1 (20.0) | 0.40 [0.04–3.66] | 0.42 | 41 (29.5) | 10 (62.5) | 3.6 [1.14–11.43] | 0.03 |

**(C) Risk of acetaminophen and ibuprofen use in subgroups of male children with asthma**

| Antipyretics | Without asthma exacerbation in the previous year | | | | Asthma exacerbation in the previous year | | | |
|---|---|---|---|---|---|---|---|---|
| | Asthma exacerbation | | Adjusted OR | | Asthma exacerbation | | Adjusted OR | |
| | No (%) | Yes (%) | (95% CI) | P-value | No (%) | Yes (%) | (95% CI) | P-value |
| Acetaminophen | 302 (56.8) | 7 (38.9) | Reference | | 34 (70.8) | 10 (55.6) | Reference | |
| Ibuprofen | 230 (43.2) | 11 (61.1) | 1.93 [0.72–5.14] | 0.19 | 14 (29.2) | 8 (44.4) | 2.49 [0.76–8.18] | 0.13 |

| Antipyretics | Non-use of systemic corticosteroids in the previous year | | | | Use of systemic corticosteroids in the previous year | | | |
|---|---|---|---|---|---|---|---|---|
| | Asthma exacerbation | | Adjusted OR | | Asthma exacerbation | | Adjusted OR | |
| | No (%) | Yes (%) | (95% CI) | P-value | No (%) | Yes (%) | (95% CI) | P-value |
| Acetaminophen | 188 (55.3) | 2 (25.0) | Reference | | 148 (61.7) | 15 (53.6) | Reference | |
| Ibuprofen | 152 (44.7) | 6 (75.0) | 3.67 [0.72–18.8] | 0.12 | 92 (38.3) | 13 (46.4) | 1.86 [0.78–4.43] | 0.16 |

**Notes:**
Model adjusted for age, sex, and patient status in the previous year (AE and use of systemic corticosteroids).
OR, odds ratios; CI, confidence intervals.

($P = 0.55$), whereas among those receiving systemic corticosteroids in an ER visit or hospitalization, the risk was higher in the ibuprofen group than in the acetaminophen group (OR = 3.15, 95% CI [1.32–3.52], $P = 0.01$).

**Table 4 Analysis of patients with systemic corticosteroid prescription in the previous year.**

| | Only in OPD | | | | In ER/hospitalization (with or without OPD) | | | |
| | Asthma exacerbation | | Adjusted OR | | Asthma exacerbation | | Adjusted OR | |
| | No (%) | Yes (%) | (95% CI) | P-value | No (%) | Yes (%) | (95% CI) | P-value |
|---|---|---|---|---|---|---|---|---|
| Acetaminophen | 190 (96.9) | 6 (3.1) | Reference | | 56 (78.9) | 15 (21.1) | Reference | |
| Ibuprofen | 113 (95.0) | 6 (5.0) | 1.44 [0.43–4.83] | 0.55 | 20 (54.1) | 17 (45.9) | 3.15 [1.32–7.52] | 0.01 |

**Notes:**

Model adjusted for age, sex, and patient status in the previous year (AE and use of systemic corticosteroids).

OR, odds ratios; CI, confidence intervals; OPD, outpatient department.

## DISCUSSION

In this study, we identified children aged 1–5 years with asthma who received acetaminophen or ibuprofen prescription alone for 52 weeks from the NHIRD to compare the rate of AE and analyze the potential risks. We identified AE on the basis of an ER visit or hospitalization due to asthma. The ibuprofen group exhibited a higher risk of an ER visit or hospitalization for asthma and had significantly higher cumulative AEs over 52 weeks in our study. Among children who had an ER visit or hospitalization in the previous year, the ibuprofen group carried a higher risk of severe AE than the acetaminophen group. This difference was not evident in children who had no ER visit or hospitalization for asthma in the previous year, regardless of whether they used systemic corticosteroids.

We focused on antipyretics because they are the most frequently prescribed pediatric drugs (*Vernacchio et al., 2009*). From the data in the NHIRD, acetaminophen and ibuprofen account for 98.62% of antipyretic prescriptions in children 1–5 years old (R. Chung, Y. Huang, Y. Chen, L. Fu, C. Lin, 2005, unpublished data), and respiratory infection is the most common condition associated with antipyretic use (*Sordillo et al., 2015*). The impacts of antipyretic prescription and respiratory infection on asthma control are a concern. However, it is difficult and almost impossible not to use antipyretics when a child has a fever, and there are ethical concerns related to designing a clinical study to compare asthma control between children using and not using antipyretics under such a condition. Studies comparing the effects of different antipyretics on asthma control may provide valuable information for medical decisions in clinical practice.

A few randomized, double-blind trials have compared acetaminophen and ibuprofen use for asthma control (*Lesko et al., 2002*; *Matok et al., 2017*; *Sheehan et al., 2016*). The Boston University Fever Study evaluated the rates of hospitalization and outpatient visits for asthma among children who had asthma and a febrile illness within 4 weeks after receiving acetaminophen or ibuprofen. Compared with the acetaminophen group, the ibuprofen group had fewer outpatient visits but no difference in asthma-related hospitalization (*Lesko et al., 2002*). The Acetaminophen vs. Ibuprofen in Children with Asthma (AVICA) trial was a long-term study that followed children aged 1–5 years for 52 weeks; it found no difference in AE, as evaluated by the intake of systemic corticosteroids, in groups receiving acetaminophen or ibuprofen alone (*Sheehan et al., 2016*). *Riley et al. (2016)* are recruiting for a randomized controlled trial of acetaminophen

vs. ibuprofen use during infancy and its association with the risk of asthma. In the present NHIRD study, which also examined children with asthma taking ibuprofen or acetaminophen over 52 weeks in 2010, no difference in AE between the two groups was found with respect to systemic corticosteroid use. However, we found that the ibuprofen group had a higher rate of asthma ER visits or hospitalization, which frequently occurred within 4 weeks of taking the antipyretic during our study period.

Sordillo et al. (2015) demonstrated that RTIs might confound the relationship between infant antipyretic use and early childhood asthma. Other studies adjusting for RTIs when analyzing early-life acetaminophen use and asthma have showed similar results (Lowe et al., 2010; Schnabel & Heinrich, 2010). The AVICA trial found that RTIs carry higher risks of antipyretic use and AE (Sheehan et al., 2016). The Boston University Fever Study (Lesko et al., 2002) also considered RTIs. In our study, the frequencies of RTI in the study period and the previous year were similar between the acetaminophen and ibuprofen groups.

Factors contributing to AE include poor asthma control, severe exacerbation in the past year, viral infection, allergen exposure, virus–allergen interaction, smoking, and air pollution. Among these factors, severe exacerbation in the past year is the strongest independent factor (Fu & Tsai, 2014; Fuhlbrigge et al., 2012). Our study focused on severe AE as indicated by asthma-related ER visits or hospitalization in the previous year, which was not analyzed in previous studies. There is no "urgent OPD care" in the current health care system in Taiwan; therefore, we included only ER visits and hospitalization. Concerning the need for systemic corticosteroids, the AVICA trial had four scenarios for initiating systemic corticosteroids—poor asthma control, an unscheduled visit to the OPD or ER due to asthma, hospitalization, and physicians' discretion. The present study could identify systemic corticosteroid use only in OPD visits, ER visits, and hospitalization.

Several studies have investigated the effects of acetaminophen on lung function. Ioannides et al. (2014) showed no significant effect of 12-week-long acetaminophen intake (1g bid) on bronchial hyperresponsiveness in adults with asthma. Soferman et al. (2013) administered children with asthma 15 mg/kg acetaminophen and then performed spirometry and measured fractional exhaled nitric oxide 60 min later. The results were not different from those of the control group. Matok et al. (2017) conducted a cross-sectional study on the association between antipyretics and wheezing in children with febrile illness; in multivariate analysis, they found that ibuprofen was associated with a lower risk of wheezing.

Several biological mechanisms underlying acetaminophen-induced asthma development and acute attack have been proposed (Henderson & Shaheen, 2013). In vitro and animal studies have revealed that N-acetyl-p-benzoquinone imine, one of the metabolites of acetaminophen, leads to the depletion of glutathione, which acts as an important antioxidant in the airways; further epithelial damage and airway inflammation may occur. Acetaminophen may also lead to preferential Th2 cytokine responses as it decreases intracellular glutathione levels and modulates cytokine production in human alveolar macrophages and type II pneumocytes in vitro (Dimova et al., 2005). Furthermore, aspirin

and most non-steroid anti-inflammatory drugs, including ibuprofen, inhibit cyclo-oxygenase-1 activity. This may lead to dysregulation of arachidonic acid metabolism and eventually the clinical symptoms of non-steroid anti-inflammatory drug-exacerbated respiratory diseases (NERDs), (*Simon, Dazy & Waldram, 2015*) previously known as aspirin-exacerbated respiratory diseases. A meta-analysis for NERD showed its prevalence to be 9% in adults (*Morales et al., 2015*). Studies have found ROS-related and TL4 gene polymorphisms to be significantly associated with the effects of acetaminophen use in asthma (*Kang et al., 2013*; *Lee et al., 2014*). In this study, we found a stronger effect of ibuprofen in the children with asthma.

This study has some limitations. In clinical practices, there are no golden standards in diagnosing asthma among patients under 5 years old except for recurrent symptoms and physicians' judgement; however, the methodology used in this study was similar and even more rigorous than previous published literatures (*Hung et al., 2014*; *Wang et al., 2013*, *2017*). There was a lack of data on the immunoglobulin E level, total eosinophil count, and aeroallergen sensitization; however, no study has shown that the atopic condition influences the effect of antipyretics on asthma control. This study is a retrospective database analysis, and we could not be clear if the patients did take the antipyretics or evaluate the cumulative antipyretic dose; however, the AVICA trial found that the cumulative antipyretic dose is not related to asthma control (*Sheehan et al., 2016*). This study focused on children with persistent asthma aged 1–5 years old, so the results may not be applied to other age groups or children with intermittent to mild asthma.

## CONCLUSIONS

In conclusion, this study using the NHIRD showed that an ER visit or hospitalization for asthma in the previous year is a major risk factor for AE. Ibuprofen carries a higher risk of AE than acetaminophen in children with asthma, especially for those who have severe AE with an ER visit or hospitalization, or systemic corticosteroid prescription in the previous year. We suggest that studies analyzing the risk of antipyretics in asthma control consider severe AE in the previous year. In clinical practice, it is difficult to avoid prescribing antipyretics to febrile children, but pediatricians should use antipyretics after full evaluation of children with asthma and be cautious of the risk of AE.

## ABBREVIATIONS

| | |
|---|---|
| AE | asthma exacerbation |
| ER | hemergency room |
| NHIRD | National Health Insurance Research Database |
| OR | odds ratio |
| OPD | outpatient department |
| CI | confidence interval |
| RTI | respiratory tract infection |
| ICS | inhaled corticosteroid |
| LTRA | leukotriene receptor antagonist. |

## ACKNOWLEDGEMENTS

This study is based in part on data from the National Health Insurance Research Database provided by the National Health Insurance Administration, Ministry of Health and Welfare and managed by National Health Research Institutes (Registered number 101095, 102148). The interpretation and conclusions contained herein do not represent those of National Health Insurance Administration, Ministry of Health and Welfare or National Health Research Institutes.

The authors would like to thank the Healthcare Service Research Center of Taichung Veterans General Hospital for statistical support.

This manuscript was edited by Wallace Academic Editing.

### Funding

This study was supported by grants from Taichung Veterans General Hospital, Taiwan (TCVGH-NHRI10603, TCVGH-1067310C, TCVGH-FCU1068205, TCVGH-YM1060201, TCVGH-VTA106PREM1, TCVGH-1076504B). There was no additional external funding received for this study. The funders had no role in study design, data collection and analysis, decision to publish, or preparation of the manuscript.

### Grant Disclosures

The following grant information was disclosed by the authors:
Taichung Veterans General Hospital, Taiwan: TCVGH-NHRI10603, TCVGH-1067310C, TCVGH-FCU1068205, TCVGH-YM1060201, TCVGH-VTA106PREM1, TCVGH-1076504B.

### Competing Interests

The authors declare that they have no competing interests.

### Author Contributions

- Lin-Shien Fu conceived and designed the experiments, prepared figures and/or tables, authored or reviewed drafts of the paper, approved the final draft.
- Che-Chen Lin performed the experiments, analyzed the data, contributed reagents/ materials/analysis tools, prepared figures and/or tables, authored or reviewed drafts of the paper, approved the final draft.
- Chia-Yi Wei analyzed the data, prepared figures and/or tables, authored or reviewed drafts of the paper, approved the final draft.
- Ching-Heng Lin conceived and designed the experiments, performed the experiments, analyzed the data, contributed reagents/materials/analysis tools, prepared figures and/or tables, authored or reviewed drafts of the paper, approved the final draft.
- Yung-Chieh Huang conceived and designed the experiments, prepared figures and/or tables, authored or reviewed drafts of the paper, approved the final draft.

## Human Ethics

The following information was supplied relating to ethical approvals (i.e., approving body and any reference numbers):

This study was approved by the institutional review board of Taichung Veterans General Hospital (No. CE16220B).

## Data Availability

Raw data for this work was obtained by application from the National Health Insurance Research Database, Taiwan (http://nhird.nhri.org.tw/en/index.html) and may not be shared according to the Database's rules governing use. Access to the data used in this study may be obtained by citizens of Taiwan who fulfill the requirements of conducting research projects.

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
