# Peer review of "Risk of acute exacerbation between acetaminophen and ibuprofen in children with asthma"

_PeerJ, doi:10.7717/peerj.6760_

## Round 0.1 · original submission · Major Revisions

Dear author,

The manuscript is very interesting however there are some concerns that need to be addressed, such as the dosage of the drugs used. Hope you will be prepared to make the necessary changes and address all reviewers queries.

·

Basic reporting

The data reported in the manuscript are improper formatting (i.e. puntuaction).
Introduction is clear and well written.
Literature is well referenced.
Methods are adeguately described.
In the table 2 the legend (a, b, c) is not clear.

Experimental design

The experimental design is clear

Validity of the findings

No comment

Additional comments

The manuscript entitled ” Risk of acute exacerbation between acetaminophen and ibuprofen in children with asthma” by Yung-Chieh Huang et al., is a retrospective analysis regarding the acetaminophen or ibuprofen use on risk of acute exacerbation in children with asthma.
The study is very useful to answer the important practical question from clinicians regarding which medication to use, acetaminophen or ibuprofen, in children with asthma that necessitate a treatment with an antipyretic, analgesic medication.

However, I have some comments.
1) The study to be appropriate and specifically designed should include the dosage of acetominophen and ibuprofen.

2) The authors should explain the adherence to asthma-controller medications among the patients.

3) Is there a history of aspirin sensitivity among the patients? Since there is cross sensitivity to all classes of NSAIDs (Kowalski ML et al., Allergy. 2011 J;66:818-29), this aspect could be considered .

4) The authors do not perform a randomized study. Therefore they cannot rule out the possibility that the asthma exacerbations are caused by the respiratory infections themselves.

5) The study intends to retroprospectively evaluate the effect of the use of acetaminophen versus ibuprofen in young children with persistent asthma. The authors should clarify that their study may not generalizable to other age groups or children with intermittent to mild asthma.

·

Basic reporting

In this study the authors compare the risk of asthma exacerbation between1-year use of acetaminophen and 1-year use of ibuprofen in children aged 1–5 years with persistent asthma, focusing on the risk factors, including severe asthma exacerbation in the past.
The paper is widely interesting since raise imporatnt questions regarding pediatricians prescriptions of the use of antipyretics among children with asthma withouth a full evaluation of the risk.

Experimental design

This study is a retrospective database analysis where the authors correlate asthma exacerbation and the therapeutic use of acetaminophen or ibuprofen. Asthma exacerbation is assessed as an acute or sub-acute episode of a progressive increase in asthma symptoms,associated with airflow obstruction. Althought factors contributing to asthma exacerbation include poor asthma control, severe exacerbation in the past year, viral infection, allergen exposure, virus–allergen interaction, smoking, and air pollution the authors include as clinical parameters only emergency room visits and hospitalization

Validity of the findings

They conclude thet Ibuprofen carries a higher risk of asthma exacerbation than acetaminophen in children with asthma, especially for those who have severe AE with an ER visit or hospitalization, or systemic corticosteroid prescription in the previous year.

The data are very interesting but the results difficult to follow how they have been reported.
It is know and also the authors confirm thet asthma incidence is sex related ( e.g. in the study the numbers of male patients is doubled respect to female) and also the therapy is widely different. In order to further demonstrate their hypothesis it might be better express the final results separately in male e female patients.

Additional comments

The data and the study are interesting and of considerable therapeutic importance given the wide use of antipyretics in children, in any case considering the different incidence of the disease between the two genera, it would be interesting to present a separate data in order to highlight useful differences in therapeutic approach.

---

## Round 0.2 · accepted · Accept

Dear author, following your revision the manuscript has been improved.

·

Basic reporting

No comment

Experimental design

No comment

Validity of the findings

No comment

Additional comments

The authors made all the changes required by the reviewers.